# ADAR1 Suppresses Interferon Signaling in Gastric Cancer Cells by MicroRNA-302a-Mediated IRF9/STAT1 Regulation

**DOI:** 10.3390/ijms21176195

**Published:** 2020-08-27

**Authors:** Lushang Jiang, Min Ji Park, Charles J. Cho, Kihak Lee, Min Kyo Jung, Chan Gi Pack, Seung-Jae Myung, Suhwan Chang

**Affiliations:** 1Department of Biomedical Sciences, College of Medicine, Asan Medical Center, University of Ulsan, Seoul 05505, Korea; qq1641221879@163.com (L.J.); alswl5011@nate.com (M.J.P.); formidable1981@gmail.com (C.J.C.); rlgkr4832@naver.com (K.L.); 2Department of Convergence Medicine, College of Medicine, Asan Medical Center, University of Ulsan, Seoul 05505, Korea; jungminkyo1@gmail.com (M.K.J.); changipack@amc.seoul.kr (C.G.P.); 3Department of Gastroenterology, College of Medicine, Asan Medical Center, University of Ulsan, Seoul 05505, Korea; 4Department of Physiology, College of Medicine, Asan Medical Center, University of Ulsan, Seoul 05505, Korea

**Keywords:** RNA editing, gastric cancer, ADAR1, microRNA-302a, type I interferon, IRF9, STAT1

## Abstract

ADAR (adenosine deaminase acting on RNA) catalyzes the deamination of adenosine to generate inosine, through its binding to double-stranded RNA (dsRNA), a phenomenon known as RNA editing. One of the functions of ADAR1 is suppressing the type I interferon (IFN) response, but its mechanism in gastric cancer is not clearly understood. We analyzed changes in RNA editing and IFN signaling in ADAR1-depleted gastric cancer cells, to clarify how ADAR1 regulates IFN signaling. Interestingly, we observed a dramatic increase in the protein level of signal transducer and activator of transcription 1 (STAT1) and interferon regulatory factor 9 (IRF9) upon ADAR1 knockdown, in the absence of type I or type II IFN treatment. However, there were no changes in protein expression or localization of the mitochondrial antiviral signaling protein (MAVS) and interferon alpha and beta-receptor subunit 2 (IFNAR2), the two known mediators of IFN production. Instead, we found that miR-302a-3p binds to the untranslated region (UTR) of IRF9 and regulate its expression. The treatment of ADAR1-depleted AGS cells with an miR-302a mimic successfully restored IRF9 as well as STAT1 protein level. Hence, our results suggest that ADAR1 regulates IFN signaling in gastric cancer through the suppression of STAT1 and IRF9 via miR-302a, which is independent from the RNA editing of known IFN production pathway.

## 1. Introduction

A-to-I RNA editing is one of the major RNA modifications that occurs co-transcriptionally in the nucleus [1,2]. Interestingly, most of the editing sites are introns or UTR regions [3,4], suggesting that they play a role in splicing or expression regulation. Despite this expectation, one of the well-established functions of RNA editing is self vs. non-self-recognition of dsRNA [5,6,7]. The edited RNA is regarded as self, whereas the non-edited one, such as exogenously introduced viral RNA, is recognized as non-self so that it triggers interferon (IFN) signaling [8,9]. The RNA editing is catalyzed by either ADAR (adenosine deaminase acting on RNA) or APOBEC (apolipoprotein B mRNA editing catalytic polypeptide-like 1) enzymes [10]. Among those, ADARs are known to catalyze A-to-I editing, which is increased in various type of tumors [11] and contributes to proteomic diversity [12]. In the context of tumors, it seems that the A-to-I editing is important not only in proteomic diversity, but also in the suppression of IFN signaling, as tumors generally generate more RNA for continuous proliferation and they need to be marked as self to avoid immune recognition. Indeed, a recent report proved this concept by showing that the loss of ADAR1 increases inflammation and sensitizes tumors to programmed death-ligand 1 (PD-L1) blockade [13].

Although the function of ADAR1 in cancer has been actively investigated, the study of RNA editing in gastric cancer is in its early stage. Since two pioneering reports introduced the impact of ADAR1 in gastric cancer progression [14,15], others have reported targets including Antizyme Inhibitor 1 (AZIN1) [16], mTOR/p70S6K pathway [17], and Phosphatase And Actin Regulator 4 (PHACTR4) [18]. All these studies consistently demonstrated the oncogenic role of ADAR1 in gastric cancer progression. Until now, however, whether and how ADAR1 affects IFN signaling pathway in gastric cancer cells are not fully understood. In addition, it is unclear on which target the ADAR1 exerts its inhibitory action on IFN signaling pathway. Therefore, we sought to determine the specific target(s) where ADAR1 exerts its anti-IFN activity and to determine whether the IFN suppressive function of ADAR1 is controlled by its RNA-editing activity.

## 2. Results

### 2.1. Generation of an Inducible ADAR1 Knockdown System in AGS Gastric Cancer Cell Line

To investigate ADAR1-dependent editing targets in the IFN signaling pathway, both p110 and p150 isoforms of ADAR1 were depleted in AGS cells by the infection of an inducible lentiviral shRNA, which targets 3′-UTR of ADAR1 gene (ADAR1 KD). Both mRNA levels (Figure 1a; * *p* < 0.05) and protein levels (Figure 1b) of ADAR1 were markedly decreased by doxycycline treatment. An immunofluorescence analysis revealed that most of the ADAR1 was abundant in the nucleus of AGS cells at homeostasis, especially enriched in the nucleoli, which was clearly decreased upon knockdown (Figure 1c, bottom left panel). Stable knockdown of ADAR1 was also achieved by lentiviral shRNA expression, as confirmed by qRT-PCR (Appendix A; ** *p* < 0.01) and immunoblot (Appendix A).

### 2.2. ADAR1 Knockdown Enhances Interferon-Stimulated Gene (ISG) Transcription and Translation upon IFN Treatment in AGS Cells

Multiple reports have shown the suppressive role of ADAR1 on IFN signaling [19,20]. We, therefore, started by validating the IFN response of control and ADAR1 KD cells. In the absence of IFN stimuli, RNA levels of most ISGs were not altered or slightly decreased (Figure 2a; * *p* < 0.05; N.S., Not Significant) in ADAR1 KD cells, except for IRF9 and STAT1, which showed a significant increase (Figure 2a,b; ** *p* < 0.01, *** *p* < 0.001). As expected, IFN treatment significantly increased the RNA expression of all tested ISGs in ADAR1 KD cells (Figure 2a). We also found that the level of STAT1 mRNA was dynamically upregulated after 1–9 h of IFN treatment in ADAR1 KD cells (Figure 2b; *p* = 0.049, ANOVA test). Consistently, immunoblot confirmed that the protein levels of STAT1, *p*-STAT1, and IRF9 were further increased upon the knockdown of ADAR1 in the presence of IFN (Figure 2c, Appendix A for quantitation).

### 2.3. Increase in IRF9 and STAT1 Level upon ADAR1 Knockdown in the Absence of IFN

STAT1 and IRF9 are mediators of canonical interferon signaling, which is initiated upon the binding of IFN to the interferon receptor (IFNAR1/2). Interestingly, we found an increase in IRF9 and STAT1 protein level upon ADAR1 knockdown in AGS cells even in the absence of IFN treatment (Figure 3a, left panels). To confirm that these alterations upon ADAR1 knockdown were not due to IFN production per se, we measured secreted IFN level from control and ADAR1 KD cells by ELISA and found no IFN expression (data not shown). Further, we checked targets which were known to be acting upstream of STAT1 and IRF9 in IFN signaling pathway including interferon regulatory factor 3 (IRF3), MDA5, and Janus Kinase 1 (JAK1), but observed no expression change (Figure 3a, right panels). This result suggested that the regulation of STAT1 and IRF9 by ADAR1 occurred in an IFN-independent manner.

### 2.4. Increased IRF9, Triggered by ADAR1 Knockdown, Was Mediated by miR-302a-3p Suppression

Considering the non-canonical activation of IRF9/STAT1 (shown in Figure 3a) as well as the discrepancy between qRT-PCR (less than 2-fold change) in Figure 2b and immunoblot results in Figure 2c, we speculated that ADAR1 affects the post-transcriptional regulation of IRF9 and STAT1 protein levels. Since our previous study revealed that miR-302a-3p and miR-302-5p were the most decreased miRNAs upon ADAR1 knockdown [18], we first examined whether miR-302a-3p could post-transcriptionally regulate IRF9. Indeed, the seed sequence of miR-302a-3p matched with the 3′-UTR of IRF9 (Figure 3b). In addition, by real-time PCR, we confirmed that the knockdown of ADAR1 decreased miR-302a-3p level (Figure 3c; *** *p* < 0.001). As the mRNA and protein levels of IRF9 were increased in the ADAR1-depleted sample (Figure 2a,c), we speculated that suppression by miR-302a-3p might have been mitigated. To prove this idea, we generated wild-type (WT) and mutant IRF9 UTR reporters with a mutated seed-binding sequence of miR-302a-3p. In the luciferase assay, we observed a significant decrease in the WT reporter activity by an miR-302a-3p mimic, that was restored by mutagenesis of the seed-binding sequence (Figure 3d; * *p* < 0.05). These data indicated that miR-302a-3p suppressed IRF9 via its 3-′UTR and explained how IRF9 was increased in ADAR1-depleted cells. Consistently, the treatment of ADAR1 KD cells with the miR-302a-3p mimic attenuated the upregulation of IRF9 and STAT1 (Figure 3e, Appendix A for quantitation).

### 2.5. ADAR1 Knockdown Mediates Decrease in Mature miR-302/367 Cluster Components and Their Precursor

The miR-302a gene exists in ‘the miR-302/367 cluster’, along with other miR-302s and miR-367, located in chromosome 4 (Figure 4a). The miR-302/367 cluster shares a common promoter and its seed sequences match to IRF9 3′-UTR (Figure 4b) [21]. Hence, we further investigated whether other miR-302s are also decreased upon ADAR1 KD. Indeed, miR-302b-3p, miR-302c-3p, and miR-302d were also decreased in ADAR1 KD cells (Figure 4c; *** *p* < 0.001), suggesting that the whole miR-302/367 cluster is responsible for the altered IRF9 level in AGS cells. Further, miR-302a-5p, which is derived from precursor miR-302a and later dissociates from miR-302a-3p, also decreased upon ADAR1 knockdown (Figure 4d; *** *p* < 0.001). Finally, we questioned whether ADAR1 mediates the miRNA changes at the stage of cleavage/mature miRNA formation or the formation of its precursor miR-302a. Results of qRT-PCR revealed that the precursor miR-302a level also decreased upon ADAR1 knockdown, suggesting that ADAR1 mediates its action via the interaction with DROSHA or by regulating the transcription of miR-302/367 cluster (Figure 4e; *p* < 0.01).

We next questioned how the STAT1 protein level was dramatically increased (Figure 3a) despite the moderate elevation of its mRNA (Figure 2b). As we found the restoration of increased STAT1 protein in ADAR1 KD cells after miR-302a-3p mimic treatment (Figure 3e), we asked whether miR-302a had a similar effect on this target. miR-302a-3p did not show any interaction on STAT1 3′-UTR (data not shown). Thus, we examined if the miR-302a-5p can regulate STAT1, as miR-302-5p can weakly bind to STAT1 UTR (Appendix A) and miR-302a-5p level was decreased in ADAR1 KD cells (Figure 4d). However, a reporter assay showed no effect of miR-302a-5p on STAT1 UTR (Appendix A), implying that miR-302a-5p does not directly regulate STAT1.

### 2.6. Depletion of ADAR1 Leads to Lower Editing Rate of MAVS and IFNAR2, But Does Not Affect Protein Expression Level or Cellular Localization

We also examined whether the editing of MAVS and IFNAR2 can affect the expression of IFN, since they are in the middle of the IFN induction pathway and were shown to be edited on our previous RNA-seq data [18]. Indeed, Sanger sequencing of the two edited sites of MAVS on RNA-seq (chr20:3850513 and chr20:3850514) in AGS cells revealed 21.6% and 16.2% of editing, respectively, in control cDNA, which were decreased to less than 10% upon ADAR1 knockdown (Appendix A). However, no detectable changes in expression levels or localization of MAVS were observed by qRT-PCR (Appendix A), immunoblotting (Appendix A), or immunofluorescence analyses (Appendix A). We then treated the cells with synthetic IFN-α, but found no difference in the editing level. The editing of IFNAR2 at position 34636384 in chromosome 21 was also confirmed by Sanger sequencing and was shown to decrease upon ADAR1 knockdown (Appendix A). However, no significant change in the expression or localization of IFNAR2 protein was observed in the absence or presence of IFN (Appendix A).

### 2.7. ADAR1 Does Not Affect Two Known Mechanisms of STAT1 Protein Stabilization, But Positively Regulates IRF9 and STAT1 Expression by miR-302/367

The alteration of phenylalanine 172 to serine (F172S) in STAT1 was shown to decrease the protein level without mRNA expression changes [22]. Treatment with the proteasome inhibitor MG132 failed to rescue the decreased STAT1 level in this cell line, which suggests that the F172S mutation affects STAT1 expression independent of proteasomal degradation. Therefore, we examined whether ADAR1 induces the alteration of phenylalanine 172 to serine (F172S) in the coiled-coil domain of STAT1, but the result showed no editing on the cDNA coding phenylalanine 172 (Appendix A). Instead, we found an amino-acid-changing editing (N92D) in STAT1 (Appendix A), but the editing efficiency seemed marginal. On the other hand, recent research showed that the expression and degradation of STAT1 protein were regulated by *protein kinase R* (PKR) in a short linear motif (SLIM)-dependent pathway. So, we tested whether ADAR1 would edit and alter the protein levels of PKR, which will in turn degrade STAT1 in AGS cells. The results in Appendix A showed that the levels of PKR and Eukaryotic translation initiation factor 2A (eIF-2α) protein were unchanged in AGS cells. These data suggest that the increased level of STAT1 might involve new mechanism. Taken altogether, we summarized ADAR1′s suppressive role on IFN signaling pathway as a schematic diagram (Figure 5). In the presence of IFN, ADAR1 suppressed STAT1/STAT2 and IRF9 as expected. We found that, even in the absence of IFN, ADAR1 still suppressed IRF9 level by influencing the expression level of miR-302a-3p and possibly other components of the miR-302/367 cluster. This suggested a new mechanism by which ADAR1 suppresses the IFN signaling pathway via the modulation of miRNA.

## 3. Discussion

IRF9 is a well-known component of the interferon-stimulated gene factor 3 (ISGF3) complex, along with phosphorylated STAT1 and STAT2, which moves into the nucleus upon stimulation by IFN and binds to Interferon-sensitive response element (ISRE) to promote the transcription of more than 100 ISGs [23]. In addition, IRF9 was previously proven as a target of miR-302d [24], the miRNA of which the seed sequence is identical to that of miR-302a-3p. In fact, these miRNAs constitute the miR-302/367 cluster, which consists of nine different miRNAs co-transcribed in a polycistronic manner. The miR-302/367 cluster is known to be highly expressed in embryonic stem cells (ESCs), and not in differentiated cells as well as somatic stem cells [25]. In accordance, well-known ESC-related transcription factors (Nanog, Oct3/4, Sox2, and Rex1 regulate the activity of the miR-302/367 cluster, which may be responsible for its turn-off in development [21]. It is interesting that the AGS cell line can produce a conceivable amount of the miR-302/367 cluster components even though it is a cancer cell line derived from fully differentiated gastric epithelial cells. Indeed, AGS was the only cell line with miR-302a-3p expression among seven cancer cell lines examined from the stomach, pancreas, as well as breast. (Appendix A). Two hypotheses can be drawn from the phenomenon: (1) dedifferentiation of fully differentiated cancer cells may have occurred during carcinogenesis, which reactivated the miR-302/367 cluster activity and (2) a proportion of cancer cells may have gained the ability to transcribe the miR-302/367 cluster, probably due to the production of Oct4 or Sox2 (known to bind to the promotor region of miR-302a [26]), and later dominated.

However, it has been reported that in H9 human ESC, the miR-302a expression is decreased during differentiation. When ADAR1 was knocked-down in embryonic stem cells, the reduction was compromised [27], which is contradictory to our result showing a positive correlation between ADAR1 and miR-302a-3p. The opposite effect of ADAR1 in the cancer cell line may be its inherent property and is a point of investigation. Thus, addressing if the inherent IFN suppressive function of ADAR1 involves miR-302a-3p will reveal its another function, independent of self-renewal.

In this study, our results showed significant STAT1 overexpression in the ADAR1-knockdown AGS cell line, suggesting that STAT1 plays a tumor suppressive role and is downregulated by ADAR1-high in gastric cancer. Although the mechanism of cytokine-induced activation of STAT1, transcriptional regulation of STAT1 gene expression [28] and post-transcriptional regulation of STAT1 protein expression have been established [29], ADAR1-dependent regulation of STAT1 expression has not been shown yet. We checked possibilities of ADAR1-mediated regulation on STAT1 gene expression, such as ADAR1 editing of MAVS and IFNAR2, which are important components of the IFN production pathway [7], as well as PKR protein which can mediate the degradation of STAT1 [30]. We could not find any evidence of STAT1 expression regulation by these molecules. Although we found that the increased STAT1 protein level by ADAR1 knockdown was reversed when the miR-302a-3p mimic was treated, the effect seemed to be indirect as an miR-302a-3p-binding site was not found on STAT1 3′-UTR.

We cannot completely rule out other mechanisms that can result in the STAT1 increase in the AGS cell line by ADAR1 knockdown. For example, a recent study showed that STAT1β can enhance STAT1 function by protecting STAT1α from degradation in esophageal squamous cell carcinoma [31]. STAT1β, a naturally spliced isoform of STAT1, lacks a 38-amino acid segment that includes the conserved STAT1 S727 phosphorylation site and most of the C-terminal transactivation domain [32]. STAT1β has not been extensively studied, although one report has described that STAT1β in human B-cells is transcriptionally inactive and exerts a dominant-negative effect on STAT1α, the full-length STAT1 isoform [33]. In our experiments, we found that both STAT1β and STAT1α showed high expression in the ADAR1-knockdown AGS cell lines (Appendix A). So, it is possible that STAT1β stabilizes STAT1α in ADAR1 KD AGS cells. In addition, it is also possible that STAT1 RNA editing (A-to-I) in a novel position can cause a structural or functional change of STAT1 protein and further lead to STAT1 degradation in AGS cells. Altogether, we present here that ADAR1 suppresses IFN signaling pathway components via the miR-302/367 cluster. We speculate that a pharmacological inhibitor or ADAR1 might be useful to abrogate this suppression that possibly reactivates the IFN response in tumor microenvironment, which may recover anti-tumor immune function in gastric cancer.

## 4. Materials and Methods

### 4.1. Cell Lines

Human gastric cancer-derived AGS cells and HEK-293T human embryonic kidney cells were purchased from Korean Cell Line Bank (Seoul, Korea). AGS Cells were maintained in RPMI medium with 10% fetal bovine serum. HEK-293T cells were grown in DMEM containing 10% fetal bovine serum. All cells were cultured at 37 °C in a humidified incubator containing 5% CO_2_.

### 4.2. RNA Extraction, cDNA Synthesis, and Quantitative Real-Time PCR

Total RNA was extracted using TRIzol reagent (Invitrogen, Carlsbad, CA, USA) and cDNA was synthesized using the PrimeScript first-strand cDNA synthesis kit (Takara, Kyoto, Japan). Human glyceraldehyde-3-phosphate dehydrogenase (GAPDH) was used as an internal control. To confirm the amplification of specific transcripts, melting curve profiles were produced at the end of each PCR, and relative quantification was calculated using the 2^−ΔΔCT^ method [34]. Quantitative assessments of microRNAs (miR-302a-3p, miR-302b-3p, miR-302c-3p, and miR-302d) were performed using TRIzol-isolated RNAs following reverse transcription with PrimeScript kits (Heimbiotek, Seoul, Korea), and cDNAs were analyzed using HB miR Multi Assay Kit™ System I (Heimbiotek, Seoul, Korea) according to the manufacturer’s instructions. Human U6 was used as an internal control. Precursor miR-302a was analyzed using the Hs_mir-302a_PR_1 miScript Precursor Assay (Qiagen, Redwood City, CA, USA). Primer sequences were:

ADAR1 forward: CTGTAGAGAAACCTGATGAAGC, reverse: GCTTGGGAACAGGGAATCG;

STAT1 forward: AGGGGCCATCACATTCACAT, reverse: AGATACTTCAGGGGATTCTC;

IRF9 forward: GAGCCAGACTACTCACTGCTG, reverse: CCACTAGGATGCCCCTCTCA;

STAT2 forward: CCAGCTTTACTCGCACAGC, reverse: AGCCTTGGAATCATCACTCCC;

OASL forward: TCACCACTGTGATGGACCTG, reverse: TCGTGCCCTCTTCACGTTC;

MAVS forward: GGGGTAACTTGGCTCCTTCT, reverse: CTTCAATACCCTTCAGCGGC;

GAPDH forward: AAGAAGGTGGTGAAGCAGGC, reverse: TCCACCACCCTGTTGCTGTA.

### 4.3. Confirmation of RNA Editing by SANGER Sequencing

To confirm RNA editing found in RNA-seq, Sanger sequencing of the candidate sites was performed by PCR amplification using the KOD FX Neo Kit (Toyobo, Osaka, Japan) and purification using the FavorPrep^TM^ GEL/PCR Purification Mini Kit (FAVORGEN, Taipei, Taiwan).

### 4.4. Knockdown and Transfection

To stably knockdown ADAR1 in AGS cells (ADAR1 KD), cells were infected with a lentivirus encoding an ADAR1 shRNA, as was previously described [18]. First, HEK-293T cells were plated at a density of 1.5 × 10^5^ cells/well in a six-well plate and incubated for 24 h. Subsequently, HEK-293T cells were co-transfected with an shADAR1 plasmid (TRCN0000050788, sequence: CCGGGCCCACTGTTATCTTCACTTTCTCGAGAAAGTGAAGATAACAGTGGGCTTTTTG, 1500 ng; Sigma, St. Louis, MO, USA,) with the packaging mix of psPAX2 (700 ng) and the envelope plasmid pMD2.G (700 ng), using Lipofectamine 2000 (Invitrogen, Carlsbad, CA, USA). Viral supernatants were then harvested at 24–48 h after transfection and were used to infect AGS cells. For interferon treatment experiments, we used an inducible ADAR1 knockdown system using the shADAR1 plasmid (TRCN0000050788) cloned into pLKO Tet-on vector, which conditionally knocks down ADAR1 by doxycycline treatment at 1 μg/mL. Using this method, ADAR1 level was stably suppressed in ADAR1 KD cells even in the presence of IFN.

### 4.5. Immunoblot Analysis

AGS and AGS ADAR1 KD cell lines were lysed in RIPA buffer containing protease inhibitor cocktail tablets. All tissue extracts were stored on ice for 10 min and then centrifuged at 14,000 rpm at 4 °C for 10 min. We collected the supernatant and measured protein concentration by using the Pierce^TM^ BCA Protein Assay Kit, and equal amounts of proteins were separated by 10% sodium dodecyl sulfate (SDS)–polyacrylamide gel electrophoresis followed by immunoblot analysis. Antibodies reactive with human β-actin (1:2000), ADAR1 (1:1000), MAVS (1:1000), interferon alpha and beta receptor subunit 2 (IFNAR2) (1:1000), Janus kinase 1 (JAK1) (1:1000), IRF9 (1:1000), and STAT2 (1:1000) were purchased from Santa Cruz Biotechnology (Santa Cruz, CA, USA). Antibodies for STAT1 (1:1000), phospho-Tyr 727 STAT1 (or phospho-STAT1 or p-STAT1) (1:1000), phospho-STAT2 (1:1000), interferon regulatory factor 3 (IRF3) (1:1000), interferon induced with helicase C domain 1 (IFIH1, also known as MDA5) (1:1000), eukaryotic translation initiation factor 2 alpha kinase 2 (EIF2AK2, also known as PKR) (1:1000), phospho-PKR (1:1000), and eukaryotic translation initiation factor 2 subunit alpha (EIF2S1, also known as eIF-2α) (1:1000) were purchased from Cell Signaling (Danvers, MA, USA).

### 4.6. Plasmid DNA and Dual-Luciferase Reporter Assay

The AGS cells were seeded in 24-well plates, 24 h prior to transfection. On the following day, 100 ng of the pMIR-REPORT™ Luciferase vector (Life technologies, Carlsbad, MD, USA) with a wild-type UTR-pMIR reporter vector or a mutant UTR reporter with a mutated seed-binding sequence was co-transfected with 10 ng of TK-luc plasmid using Lipofectamine 2000 (Invitrogen, Carlsbad, CA, USA). miR-302a mimic or scramble RNA duplexes were also co-transfected (Genolution, Seoul, Korea). The cells were harvested 24–48 h later, suspended in passive lysis buffer, and the activity of the luciferase reporter was measured by the Dual-Luciferase Reporter Assay System (Promega, Madison, WI, USA), following the manufacturer’s instruction. Relative luciferase expression was determined as the ratio of firefly to Renilla luciferase activity. When miR-302a was transfected, AGS cells were plated in 6-well plates, 24 h prior to transfection. On the following day, the miRNA mimic or scrambled control was transfected using Lipofectamine 2000 (Invitrogen, Carlsbad, CA, USA) for 48 h.

### 4.7. Synthetic Type I IFN and ELISA Kit for Detection

For stimulation with type I IFN, IFN-α-2b (11105-1, PBL, Piscataway, NJ, USA) was used. A VeriKine™ Human IFN-Alpha ELISA kit was used to detect type I IFN (PBL IFN Source, Piscataway, NJ, USA) from supernatants of AGS cells, and human IFN-γ was detected by ELISA using the Quantikine™ kit (R&D Systems, Minneapolis, MN, USA).

### 4.8. Immunofluorescence

Cells were seeded on Lab-Tek 8-well chambered coverglasses (Nalgene Nunc, Buffalo Grove, IL, USA) in RPMI media and were grown for 48 h. Cells were washed in PBS and fixed with 5% paraformaldehyde in PBS at pH 7.4 for 30 min, permeabilized with 0.2% Triton X-100 for 10 min, and then incubated in a blocking solution (1% bovine serum albumin in PBS) for 30 min. Cells were then incubated for 2 h with primary antibodies, followed by 1 h of incubation with secondary antibodies and Hoechst33342 (R37605, Life Technologies, Carlsbad, MD, USA).

### 4.9. Statistics Analyses

Data are presented as mean ± standard deviation of two or three independent experiments. A comparison between two groups was performed using Student’s *t*-test. Data are presented as mean ± standard error. Differences were considered significant when *p*-values were <0.05.

## Figures and Tables

**Figure 1 ijms-21-06195-f001:**
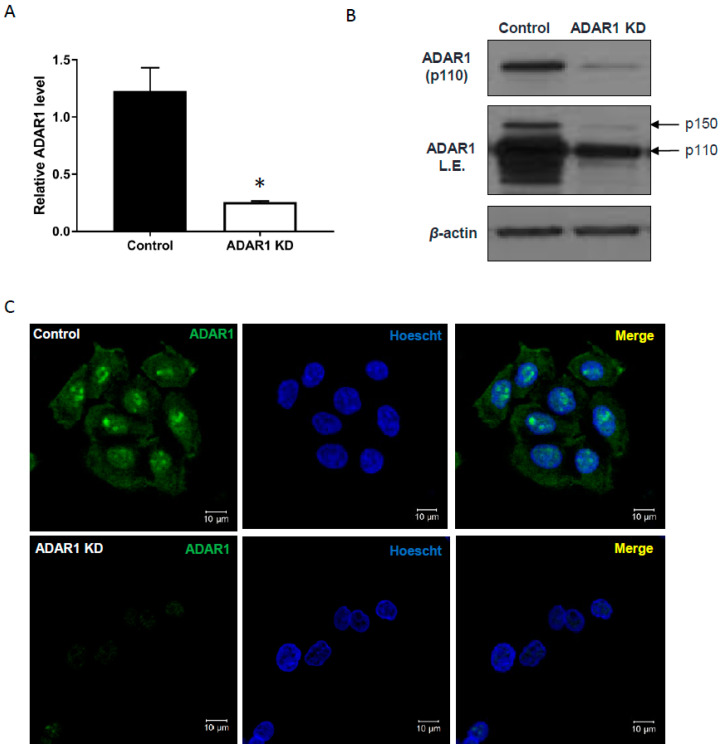
Inducible knockdown of ADAR1 in AGS gastric cancer cells. (**A**) qRT-PCR, (**B**) immunoblot, and (**C**) immunofluorescence analysis of ADAR1 expression in AGS cells infected with Tet-on shADAR1 or control (empty vector). In (**C**), green signal on the left indicates ADAR1 protein and the middle one is for Hoechst staining. Merged image is shown on right. Scale bar, 10 µM; * *p* < 0.05.

**Figure 2 ijms-21-06195-f002:**
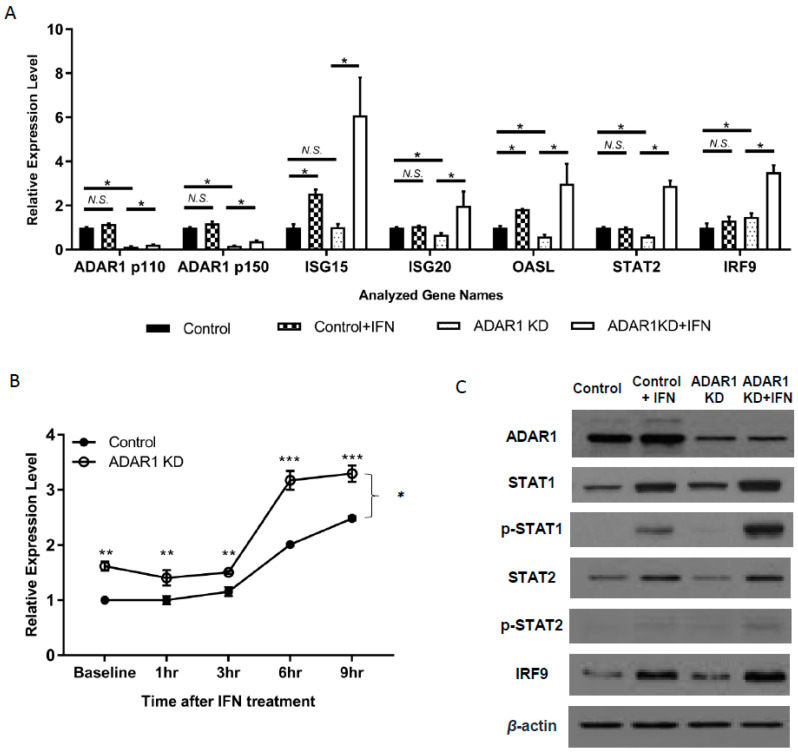
Increased expression of interferon-stimulated genes (ISGs) in ADAR1 knockdown (KD) cells after interferon (IFN) treatment. (**A**) Real-time PCR results show the expression of ISG15, ISG20, OASL, STAT2, and IRF9 by RT-PCR in AGS control and ADAR1 KD cell lines with or without IFN treatment. (**B**) Time-dependent STAT1 mRNA expression changes after IFN treatment in control and ADAR1 KD cells. (**C**) Immunoblot results of key proteins for IFN production or signaling pathway upon the knockdown of ADAR1. * *p* < 0.05, ** *p* < 0.01, *** *p* < 0.001.

**Figure 3 ijms-21-06195-f003:**
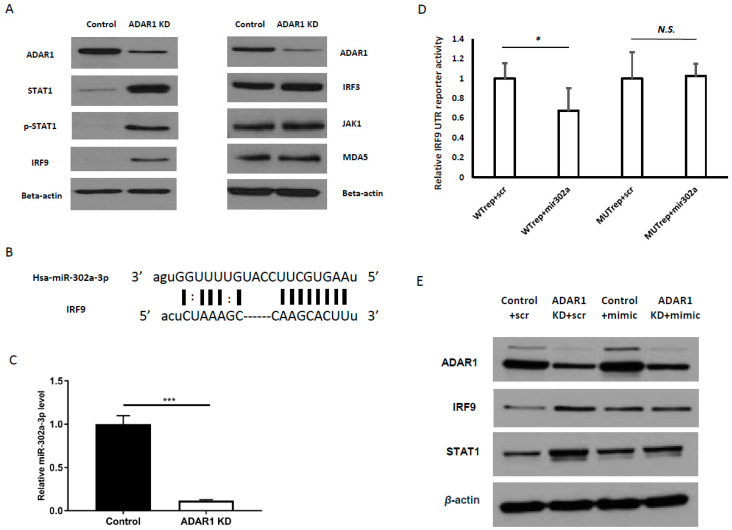
miR-302a-3p downregulation results in enhanced IRF9 expression upon ADAR1 knockdown. (**A**) Right panels: Immunoblot results showing increased STAT1, p-STAT1, and IRF9 in ADAR1 KD cells. Left panels: Immunoblot results showing unchanged protein levels of IRF3, JAK1, and MDA5, the upstream targets of STAT1 and IRF9. (**B**) Alignment between IRF9 UTR and miR-302a-3p showing seed sequence matching with the 3′-UTR of IRF9 mRNA. (**C**) Real-time PCR results of miR-302a-3p in control and ADAR1 KD cells. (**D**) Luciferase reporter assay result of wild-type IRF9 UTR reporter (WT) or UTR mutant with the seed-binding site of miR-302a-3p (MUT), co-transfected with the miR-302a-3p mimic or scrambled control. (**E**) Immunoblot images showing STAT1 and IRF9 level in control and ADAR1 KD cell lines, treated with the miR-302a-3p mimic or scrambled control. * *p* < 0.05, *** *p* < 0.001.

**Figure 4 ijms-21-06195-f004:**
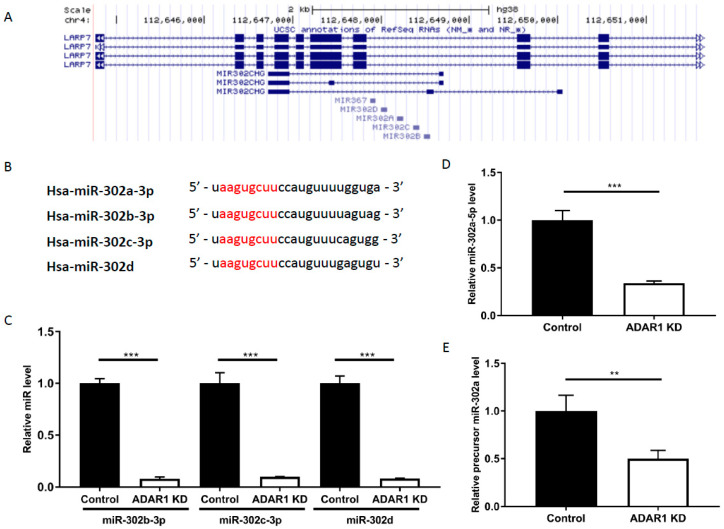
Expression of the miR-302/367 cluster is regulated upon knockdown of ADAR1. (**A**) Mapping of miR-302/367 cluster components on the UCSC genome browser (genome.ucsc.edu, GRCh37/hg19). (**B**) Comparison of sequence alignment of miR-302/367 cluster components; hsa-miR-302a-3p, hsa-miR-302b-3p, hsa-miR-302c-3p, and hsa-miR-302d share identical seed sequences (marked as red) that match the 3′-UTR of IRF9. (**C**) miR RT-PCR result showing reduced miR-302b-3p, miR-302c-3p, and miR-302d in ADAR1 KD cells compared with the control. (**D**) miR RT-PCR result showing reduced miR-302a-5p in ADAR1 KD cell line compared with the control. (**E**) Precursor miRNA RT-PCR result showing reduced precursor miR-302a level in ADAR1 KD cells compared with the control. ** *p* < 0.01, *** *p* < 0.001.

**Figure 5 ijms-21-06195-f005:**
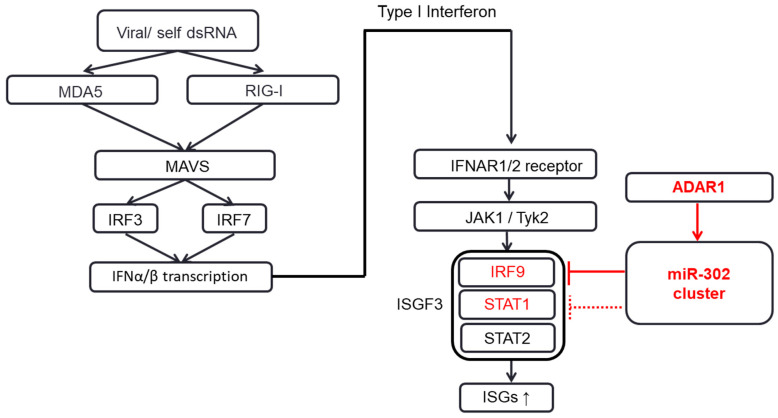
Schematic diagram of IFN signaling pathway suppressed by ADAR1 via the miR-302/367 cluster. Various intrinsic and extrinsic factors, such as viral infection, produce double-stranded RNAs (dsRNAs) which lead to the production of IFN-α and IFN-β, which in turn act on the downstream counterparts, IFNAR1/2 receptors. The downstream of IFN receptor activates JAK1/Tyk2, STAT1, STAT2, and IRF9, which subsequently form the ISGF3 complex and finally activate the interferon-stimulated genes in the nucleus, thereby enhancing the cellular immunity and related cellular functions (indicated by arrows). In the condition when miR-302/367 cluster expression is ample, knockdown of ADAR1 itself can reduce the expression of miR-302/367 cluster, thereby increasing the expression of STAT1 and IRF9 (colored in red).

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
