# Peer review of "ADAR1 Suppresses Interferon Signaling in Gastric Cancer Cells by MicroRNA-302a-Mediated IRF9/STAT1 Regulation"

_ijms, 2020, doi:10.3390/ijms21176195_

Round 1
Reviewer 1 Report
Review for Manuscript ijms-905709-peer-review-v1
General Comments: Very nicely designed study. Almost all of my comments are text changes. I have a few general comments:
1) Add P values in the Results section to help the reader.
2) Figure 1 – Report P value for t-test used with Figure 1A.
3) Figure 2 – Multiple group comparisons within each gene should be performed with a one-way ANOVA in Figure 2A.
More specific comments are listed below in order since there are no line numbers.
More Specific Comments:
Title – None
Abstract
1) Line 15 – Insert “the” before “type”
2) Line 21 – Change “for IFN” to “of IFN”
Introduction
1) Line 31 – Insert “the” before “nucleus”
2) Line 32 – Change “Despite the” to “Despite this”
3) Line 39, 40, and 42 – Change “tumor” to “tumors”
Results
1) Line 60 – Insert “the” before “nucleus”
2) Line 67 – Change “from” to “by”
3) Line 73 – Remove “that”
4) Line 77 – Insert “an” before “increase”
5) Line 82 – Remove “at the”
6) Line 89 and 93 – Change “level” to “levels”
7) Line 103 – Insert “the” before “miR-302/367 cluster”
8) Line 104 – Change “share” to “shares”
9) Line 110 – Change “mediate” to “mediates”
10) Line 117 – Change “have” to “had”
11) Line 132 – Insert “with” before “synthetic”
12) Line 140 – Insert “The” before “Proteasome”
13) Line 148 – Change “level” to “levels”
14) Line 155 – Insert “the” before “IFN”
Discussion
1) Line 201 – Insert “the” before “interferon”
2) Line 205, 206, 209, 216 – Insert “the” before “miR-302/367 cluster”
3) Line 209 – Change “clusters” to “cluster”
4) Line 215 and 220 – Change “cell” to “cells”
5) Line 223 – Reword “tempting to find out that”
6) Line 252 – Change “lead to reactivate” to “reactivates”
Methods
1) See comments about Figures 1A and 2A
Figures, Tables, and Legends – See above comments about statistical analysis.
Author Response
We sincerely appreciate for the valuable comments from the reviewer #1. Following the comments, we revised our manuscript as follows.
General Comments: Very nicely designed study. Almost all of my comments are text changes. I have a few general comments:
1) Add P values in the Results section to help the reader.
>> We added p-value for each sentence of results section.
2) Figure 1 – Report P value for t-test used with Figure 1A.
>> We thank for the critical point. We added p-value of t-test for Figure 1A and marked it in the text as well (line 64)
3) Figure 2 – Multiple group comparisons within each gene should be performed with a one-way ANOVA in Figure 2A.
>> We agree with the comment. Therefore, we did ANNOVA test for Figure 2A that gave p-value 0.049 (The raw data is in the table below). We marked it in the Figure 2A and results section (line 79).
| Annova : Single Factor | ||||||
| Summary | ||||||
| Groups | Count | Sum | Average | Variance | ||
| Con | 12 | 1.014684 | 0.084557 | 0.001059 | ||
| KD | 12 | 1.431163 | 0.119264 | 0.002274 | ||
| ANOVA | ||||||
| Source of Variation | SS | df | MS | F | P-value | F crit |
| Between Groups | 0.007227 | 1 | 0.007227 | 4.337076 | 0.049127 | 4.30095 |
| Within Groups | 0.036661 | 22 | 0.001666 | |||
| Total | 0.043888 | 23 | ||||
For other specific comments, we changed all the corrections as the reviewer pointed out. We marked these changes by Trackchange function in the revised manuscript, so please check it.
Again, we appreciate the helpful comments form the reviewer #1.

Reviewer 2 Report
I congratulate the Authors on the manuscript deciphering ADAR1-dependent IFN signaling in gastric cancer. The work significantly contributes the our understanding the role of RNA editing in this pathway. The Authors performed a series of well-designed experiments which led to a conclusion that ADAR1 suppresses IFN signaling pathway components via miR-302/367 cluster.
The manuscript has minor linguistic flows which require corrections. Below are only a few examples and the whole text should be carefully verified:
45 editing in gastric cancer is in its early stage. Since two pioneering report - should be “reports”
107 decreased in ADAR1 KD cells (Fig. 4c), suggesting that whole miR-302/367 cluster – should be “the whole”
141 to rescue decreased in STAT1 level in this cell line -should be “decrease in STAT1 level” or “ decreased STAT1”
153 We found, even in the absence of IFN, ADAR1 still suppresses IRF9 – should be “we found that even in the absence of IFN”
234 But we could not find any evidences of STAT1 – should be “any evidence”
Author Response
We really appreciate for the valuable comments from reviewer #2. We have edited our manuscript as follows.
45 editing in gastric cancer is in its early stage. Since two pioneering report - should be “reports” > We changed report to reports
107 decreased in ADAR1 KD cells (Fig. 4c), suggesting that whole miR-302/367 cluster – should be “the whole” > we changed "that" to "the"
141 to rescue decreased in STAT1 level in this cell line -should be “decrease in STAT1 level” or “ decreased STAT1”
> We deleted "in" so that it became "decreased STAT1"
153 We found, even in the absence of IFN, ADAR1 still suppresses IRF9 – should be “we found that even in the absence of IFN”
> We changed it to “we found that even in the absence of IFN”, as the reviewer suggested
234 But we could not find any evidences of STAT1 – should be “any evidence” > We deleted "s" in evidences
Again, we thank to the reviewer for helpful comments.
